# Acute Kidney Injury in Hospitalized Patients with COVID-19: Risk Factors and Serum Biomarkers

**DOI:** 10.3390/biomedicines11051246

**Published:** 2023-04-23

**Authors:** Anastasia Shchepalina, Natalia Chebotareva, Larissa Akulkina, Mikhail Brovko, Viktoria Sholomova, Tatiana Androsova, Yulia Korotchaeva, Diana Kalmykova, Elena Tanaschuk, Marina Taranova, Marina Lebedeva, Vladimir Beketov, Sergey Moiseev

**Affiliations:** 1Tareev Clinic of Internal Diseases, Department of Internal Diseases, Occupational Diseases and Reumatology, Sechenov First Moscow State Medical University, 119435 Moscow, Russia; 2Faculty of Fundamental Medicine, Lomonosov Moscow State University, Build. 1, 119991 Moscow, Russia; 3Department of Nursing, Sechenov First Moscow State Medical University, 119435 Moscow, Russia

**Keywords:** COVID-19, renal function, biomarkers

## Abstract

Background. AKI is one of the COVID-19 complications with high prognostic significance. In our research, we studied the prognostic role of several biomarkers that could help us understand AKI pathogenesis in patients with COVID-19. Methods. We evaluated the medical data of 500 patients hospitalized with COVID-19 in Tareev Clinic from 5 October 2020 to 1 March 2022. The diagnosis of COVID-19 was confirmed with positive RNA PCR in nasopharyngeal swabs and/or typical radiological findings on CT scans. Kidney function was assessed in accordance with KDIGO criteria. In the selected 89 patients, we evaluated serum levels of angiopoetin-1, KIM-1, MAC, and neutrophil elastase 2 and their prognostic significance. Results. The incidence of AKI in our study was 38%. The main risk factors for kidney injury were male sex, cardiovascular diseases, and chronic kidney disease. High serum angiopoetin-1 levels and a decrease in blood lymphocyte count and fibrinogen level also increased the risk of AKI. Conclusions. AKI is an independent risk factor for death in patients with COVID-19. We propose the prognostic model of AKI development, which includes the combination of serum levels of angiopoetin-1 and KIM-1 on admission. Our model can help to prevent AKI development in patients with coronavirus disease.

## 1. Introduction

In March 2020, the WHO announced a COVID-19 pandemic caused by the novel coronavirus SARS-CoV-2. 

Clinical manifestations of the infection range from an asymptomatic course to severe pneumonia, requiring mechanical ventilation and associated with high mortality [1]. One of the serious complications of COVID-19 is acute kidney injury (AKI), which also determines the prognosis of the disease [2].

According to several researchers, the incidence of AKI in hospitalized patients varies from 0.5% to 39% [3,4,5]. Earlier studies have suggested an important impact of cytokine storm and acute inflammatory response to acute kidney injury in COVID-19 [6]. However, the use of glucocorticosteroids and immunobiological therapy did not significantly affect the incidence of AKI, and the leading mechanisms of renal damage are still unknown [7]. The direct cytopathic effect of the virus on the renal tubular epithelium, hyperinflammation factors (“cytokine storm”), endothelial dysfunction, activation of neutrophil traps and immunothrombosis, and the contribution of various concomitant diseases, drugs, and other factors are discussed [8,9,10]. 

The aim of our study was to assess the frequency and risk factors for AKI, including those revealing the underlying pathological mechanisms of kidney damage in COVID-19. We examined the serum concentration of endothelial dysfunction marker (angiopoietin-1), marker of NETosis (neutrophil elastase 2), kidney injury molecule-1 (KIM-1), which is also the receptor for SARS-CoV-2, and the level of the membrane-attacking complex (MAC) in hospitalized COVID-19 patients, to clarify the leading mechanisms of kidney damage.

## 2. Materials and Methods

We included 500 patients hospitalized in the E.M. Tareev Clinic (Sechenov University) from October 2020 to March 2022. Diagnosis of COVID-19 based on specific findings on chest CT (ground glass opacities, reticular changes, consolidations, perilobular infiltrations) and/or real-time RT-PCR diagnostics detecting the presence of viral RNA of SARS-CoV-2. The study was conducted in accordance with the principles of the Declaration of Helsinki and was approved by the local ethics committee of Sechenov University. All patients signed an informed consent form prior to enrollment.

Demographic parameters (sex, age), the presence of concomitant diseases: arterial hypertension, coronary artery disease (myocardial infarction (MI) in the anamnesis), atrial fibrillation (AF), diabetes mellitus (DM), congestive heart failure (CHF), chronic kidney disease (CKD), obesity with an assessment of the body mass index; the area of lung tissue damage according to chest CT [11], blood oxygen saturation, positive or negative PCR for SARS-CoV-2 were assessed on admission, as well as the need and type of respiratory support, therapy that patient received before the admission and during hospitalization, the dynamics of laboratory markers; the level of acute phase proteins: CRP, ferritin, LDH, the number of lymphocytes and peripheral blood platelets, procalciotonin, D-dimer, fibrinogen, serum creatinine, glycose, electrolytes, and spot proteinuria.

AKI was diagnosed according to the KDIGO Clinical Practice Guideline (12 AKI was defined on the basis of serum creatinine changing ≥ 26.5 μmol/L from the time of admission to peak follow-up value) [12].

A total of 89 patients were selected from the main cohort to assess additional risk factors for AKI in COVID-19. The characteristics of patients in this special group are presented in Appendix A. A special group of 89 patients included 47 patients who developed AKI during hospitalization and 42 patients without AKI. These subgroups were matched for sex, age, lung detection, incidence of CKD, and other comorbidities (DM, history of MI, CHF) (Appendix A).

### 2.1. The Biomarkers Levels Assessment

10 mL samples of blood samples of patients (*n* = 89) were collected at admission in dry plastic test tubes and centrifuged at room temperature for 15 min with a rotational frequency of 3000 rpm. The serum was immediately frozen and stored in a refrigerator at the temperature of −80 °C. 

We used the commercial enzyme-linked immunosorbent assay (ELISA) kits to determine the serum levels of angiopoetin-1 (SEA008Hu, «BlueGene Biotech», Shanghai, China), kidney injury molecule-1 (SEA785Hu, «BlueGene Biotech», Shanghai, China), neutrophil elastase-2 (SEA181Hu, «BlueGene Biotech», Shanghai, China) and membrane-attacking complex (E01T0508, «BlueGene Biotech», Shanghai, China).

### 2.2. Statistical Analysis

Results are presented in medians and interquartile ranges (25th; 75th percentile). The differences between the groups were analyzed by nonparametric methods, the Mann–Whitney and Kruskal–Wallis U-test. We used the χ^2^ and Fisher tests to compare the qualitative variables. To determine the correlation relationships between the variables, we used the Spearman correlation (rs). 

The odds ratio of AKI and 95% CI were calculated using unifactorial and multifactorial logistic regression, and the hazard ratio (95% CI) of the death risk depending on the stage of AKI was performed using unifactorial Cox regression. The unifactorial logistic regression model included demographic parameters (sex, age), comorbidities (arterial hypertension, DM, CKD, CHF, AF, MI in anamnesis), clinical parameters (oxygen saturation upon admission, the percentage of lung parenchyma affected by COVID-19 lesions, types of respiratory support (absent, transnasal humidified rapid oxygen insufflation, non-invasive lung ventilation, invasive lung ventilation), laboratory markers (maximum serum levels of D-dimer, CRP, ferritin, fibrinogen, LDH, procalcitonin, potassium and sodium, the presence of proteinuria, lymphocytes, and platelets count), markers of endothelial dysfunction (angiopoetin-1), NETosis (neutrophil elastase 2), inflammation (kidney injury molecule-1), and complement activation (membrane-attacking complex). Therapy during hospitalization (immunobiological drugs: tocilizumab, olokizumab, levilimab; GCS: prednisone, dexamethasone; azithromycin, hydroxychloroquine, antibiotics, inotropes, diuretics) was included in the analysis. The results were considered statistically significant with *p* < 0.05.

Statistical analysis was performed using the IBM SPSS v.23.0 program (SPSS: An IBM Company, Armonk, NY, USA).

## 3. Results

### 3.1. Subject Characteristics

In the examined cohort (*n* = 500), AKI was developed in 190 (38%) patients. In the majority of them (79.5%), AKI stage 1, in 13% AKI stage 2, in 14 patients (7.4%), AKI stage 3 was developed (Table 1).

In the AKI group, older age and males were prevalent. Arterial hypertension, CKD stages 3–4, AF, coronary artery disease, and 3–4 functional classes of CHF, according to NYHA, were more frequent comorbidities in the AKI group than in the group with no AKI. Patients in the AKI group had significantly lower oxygen saturation on admission and more severe pulmonary lesions according to chest CT compared with the group without AKI. These groups also differed in most laboratory parameters of inflammation and blood-clotting activation (Appendix A). Patients with AKI were significantly more likely to receive antibiotic therapy and immunobiological therapy than patients with no AKI. Patients in this group were more likely to be transferred to the ICU, needed respiratory and inotropic support, and had a longer length of hospital stay. Mortality in patients with AKI was also significantly higher compared to patients with no AKI (35.3% vs. 5.5%, *p* < 0.001) (Table 1).

The risk of death increased 3.7 times [95% CI 2.1–6.6], *p* = 0.001 in the AKI group; 1.4 times [95% CI 0.9–2.2], *p* = 0.119 in AKI stage 1; 2.2 times [95% CI 1.3–3.8], *p* = 0.005 in AKI stage 2; 2.1 times [95% CI 1.1–3.9], *p* = 0.02 in AKI stage 3. Since the risk of death was not significantly different in AKI stages 2 and 3, these two subgroups were combined for further analysis.

The main cohort and the special group were comparable in sex, age, percent of lung parenchyma involvement, incidence of CKD, and other comorbidities. There were no significant differences in the main laboratory parameters at admission (D-dimer, leukocytes, lymphocytes, ferritin, glucose, LDH, fibrinogen, proteinuria). The median time from blood sampling to the development of AKI was 6 [1; 11.3] days.

### 3.2. Serum Angiopoetin-1, KIM-1, Neutrophil Elastase 2, MAC Levels in Patients with COVID-19

AKI developed in patients with a significantly higher level of angiopoietin-1 at admission compared to the patients with no AKI (Figure 1). There was a positive correlation of the serum angiopoietin-1 level with the peak levels of inflammation markers CRP and LDH (Rs = 0.258, *p* = 0.024 and Rs = 0.334, *p* = 0.003, respectively), as well as with the level of D-dimer (Rs = 0.338, *p* = 0.003), and negative correlation with the lymphocyte (Rs = −0.338, *p* = 0.003) and platelet counts (Rs = −0.411, *p* = 0.0001) and the level of fibrinogen (Rs = −0.229, *p* = 0.047).

KIM-1 serum levels on admission were lower in patients who developed AKI (Figure 2). The level of another factor, MAC, in the blood serum at admission did not differ significantly in patients with AKI and with no AKI (Figure 3). A significant increase in neutrophil elastase 2 was detected in patients with more severe AKI stages 2–3 (Figure 4). There were significant correlations of the neutrophil elastase 2 level with markers of inflammation: CRP (Rs = 0.254, *p* = 0.016), LDH (Rs = 0.255, *p* = 0.016), fibrinogen levels (Rs = 0.351, *p* = 0.045), and proteinuria (Rs = 0.306, *p* = 0.046).

### 3.3. Uni- and Multifactorial Regression Analysis

According to unifactorial logistic regression analysis, the male sex increased the risk of AKI in patients 1.7 times [95% CI 1.2–2.5, *p* = 0.003]. Among the concomitant diseases, the AKI risk was significantly increased by cardiovascular diseases: arterial hypertension, AF, coronary artery disease with a history of MI, and CHF of 3–4 functional classes. A history of CKD increased the risk of AKI by 1.8 times [95% CI 1.2–2.7, *p* = 0.008].

Among the laboratory parameters, an increased risk of AKI was associated with proteinuria, the severity of lymphopenia, high D-dimer levels, and a decrease in fibrinogen levels and platelet counts. Levels of acute phase proteins (CRP, ferritin) did not contribute to the increased risk of AKI (Table 2). Most medications in the hospital affect the risk of kidney damage. In contrast, metformin reduced the risk of AKI 1.6 times [OR 0.6 95% CI 0.3–0.9), *p* = 0.022] (Table 2) 

In a multifactorial logistic regression model, male sex, detection of proteinuria, systemic hypoxia associated with respiratory and heart failure, lymphopenia, and a decrease in fibrinogen levels had an independent effect on the risk of AKI. Moreover, an increased risk of AKI was associated with nephrotoxic antibiotics (OR 1.9 [95% CI 1.1–3.3]) (amikacin, vancomycin). In contrast, treatment with metformin reduced the risk of AKI by 2.5 times (Table 2).

### 3.4. ROC Analysis

In the ROC analysis of the risk of AKI, we noted a moderate value of each individual serum biomarker: angiopoietin-1, neutrophil elastase 2, and KIM-1 (Table 3).

In order to increase the predictive value of these factors in AKI risk, the predicted value of the markers’ combinations was determined. Among the possible options, the model including angiopoietin-1 and KIM-1 (AUC 0.833 [95% CI 0.7–0.9], *p* = 0.001, sensitivity 83.3%, specificity 76.9%) had the highest significance in determining the risk of AKI. The addition of neutrophil elastase 2 did not significantly change the AUC (0.808 [95% CI 0.6–0.9], *p* = 0.003) (Figure 5). 

## 4. Discussion

The incidence of AKI in our cohort of patients hospitalized with COVID-19 was 38%. AKI increased the risk of death by 3.7 times, which is consistent with other studies. In the research of Chan L et al., the incidence of AKI was 43.5% [13]. According to Feng X et al., acute kidney damage was observed in 30.7% of patients with COVID-19 [14]. In our cohort, AKI stage 1 increased the risk of an adverse outcome by 1.4 times, while severe kidney damage (stages 2–3) increased the risk of an unfavorable outcome by 5.4 times. These data are in agreement with the data of Cheng Y et al., who showed that AKI stage 1, stage 2, and stage 3 increased the risk of death by 1.9, 3.5, and 4.7 times, respectively [15].

Cardiovascular diseases (arterial hypertension, atrial fibrillation, CHF), as well as CKD with a decrease in GFR of less than 60 mL/min, were the comorbidities that influenced the renal injury risk in our cohort. Similar results were obtained by other authors. In the work of Chan L et al., CHF was also identified as one of the prognosis-determining comorbidities in the group of patients with AKI (13% of patients) [13]. In the other prospective study, Sullivan M et al. found that the patient’s history of CKD increased the risk of developing AKI by 1.6 times [16]. According to Hirsch JS et al., among the concomitant diseases, the independent risk factors for the development of AKI were arterial hypertension and diabetes mellitus [4].

Among the comorbidities, heart failure had an independent effect on the risk of AKI in a multifactorial regression model in our cohort. The initial impairment of intrarenal hemodynamics in this category of patients increased the risk of cardiorenal syndrome and acute decline of glomerular filtration rate, as shown in a previous study by Fang et al., congestive CHF increased the risk of developing AKI by 14.8 times [17]. Kolhe N et al. showed that the presence of congestive CHF in a patient was also an independent factor of renal damage and AKI by 1.7 times [18].

Although many risk factors for AKI are widely discussed, the underlying mechanisms of renal damage are not fully understood. This is also evidenced by the maintenance of the previous incidence of AKI, despite the use of corticosteroids and immunobiological drugs to suppress the systemic inflammatory response.

In our study, laboratory parameters for assessment were chosen in accordance with many studies dedicated to the evaluation of the severity and prognosis of the novel coronavirus infection [19]. Ng et al. discuss markers of hyperinflammation in patients with COVID-19 and AKI and refer to secondary hemophagocytic lymphohistiocytosis. Cardinal markers of this syndrome are IL-2,6,7, ferritin, and some others, which are associated with COVID-19 severity and AKI incidence [20]. Moreover, CRP, LDH, ferritin, fibrinogen, and D-dimer levels were evaluated by Yildirim et al., Serum levels of these biomarkers were significantly higher in patients with AKI than in patients with preserved renal function [21].

According to our study, independent risk factors for AKI in patients hospitalized with COVID-19 were male sex, activation of endothelial dysfunction, and mechanisms of immunothrombosis, as well as respiratory and circulatory hypoxia due to severe forms of CHF. Markers of inflammation have not demonstrated a significant impact on AKI development. The earliest mechanism of renal damage is endothelial dysfunction. Angiopoietin-1 is involved in blood vessel differentiation, growth, and remodeling [22]. In the acute phase of inflammation, it is involved in the regulation of the vascular wall tone and permeability. In our study, the serum concentration of angiopoietin-1 at admission was significantly higher in patients who had AKI at follow-up than in patients with no AKI during the in-hospital stay. In addition, angiopoietin-1 was one of the risk factors for AKI, which showed its significance in the multifactorial regression model. Given the significant correlations of angiopoietin-1 with the levels of acute phase reactants, as well as the degree of lymphopenia, it is possible to assume the effect of proinflammatory cytokines and the SARS-CoV-2 itself on the endothelium with its damage and consequent endothelial dysfunction. The correlation of the level of angiopoietin-1 with the level of D-dimer, a decrease in the number of platelets, and the level of fibrinogen indicates the activation of the processes of microthrombi formation and consumption of coagulation factors following endothelial dysfunction.

Our results are consistent with the data of other authors, who revealed a high predictive value of the serum angiopoietin-1 level in relation to 28-day mortality in patients with sepsis in the ICU departments [23]. 

Based on the evidence of the important role of NETosis in kidney damage in COVID-19, we established a significantly higher baseline serum level of neutrophil elastase 2 in patients with AKI stages 2 and 3 at a late period of hospitalization. Neutrophil elastase 2 also correlated with inflammatory markers (CRP and LDH) and proteinuria. In the Henry B et al. study, neutrophil elastase 2 levels were shown to be significantly higher in patients with AKI compared to patients without AKI, and it correlated with another surrogate marker of NETosis, extracellular DNA circulating in the blood [24]. The obtained results allow us to discuss the contribution of the NETosis process to the development of severe AKI. NETosis in circulating blood neutrophils and possibly in tissue neutrophils, which infiltrate the kidneys and lungs, plays a role in prolonging inflammation, the development of a cytokine storm, and on the other hand, can cause activation of blood clotting by the internal pathway and the process of immunothrombosis [25]. The increase in neutrophil elastase 2 in our study correlated with proteinuria, which appears to have inflammatory and ischemic genesis in COVID-19 patients.

Our study showed a tendency for patients with particularly severe stages of AKI to have lower serum levels of KIM-1 on admission. The established pattern is not entirely clear and requires further investigation. In the works of other authors, KIM-1 in patients with COVID-19 was studied in the urine, and its level was significantly increased in patients who developed AKI [26]. In addition, KIM-1 in urine is a well-studied marker of tubule damage in AKI of ischemic and toxic genesis [27].

In the serum of COVID-19 patients, KIM-1 levels were assessed by Kerget B et al., but kidney damage was not studied in this study [28]. According to the authors, serum KIM-1 was elevated in patients with extremely severe novel coronavirus infection compared to patients with a moderate course. In experimental studies, it was found that KIM-1, in addition to ACE, is a receptor for SARS-CoV-2 [29]. The KIM-1 molecule is expressed in many tissues of the body, including the tubular epithelium cells in kidneys, in response to a damaging factor, for example, hypoxia, from where it can enter the bloodstream, cleaved from the extracellular domain by metalloproteinases [30]. The KIM-1 receptor is able to bind particles of the virus, “blocking” it and preventing its further spread throughout the body. We believe that in the process of active KIM-1 binding to the epitopes of the virus, its amount in the blood may temporarily decrease, for example, immediately before the development of AKI.

The role of complement activation as one of the leading mechanisms of renal damage is discussed, but in our study, the level of MAC in the blood serum on admission did not differ significantly in patients with the development of AKI and without it, which indicates that there is no direct effect of complement activation on the processes of kidney tissue damage at least at an early stage of the disease [31].

According to the ROC analysis, serum angiopoietin-1 and KIM-1 levels were the most informative combination for predicting AKI development within a week from the day of hospitalization, indicating a predominant activation of endothelial dysfunction and the virus’s consumption of KIM-1 receptor determinants.

Metformin had a protective effect on AKI and an unfavorable outcome, reducing the risk of AKI by 2.7 times and the risk of death by 3.2 times in our study. These data are consistent with data from other studies. For example, in a study of 136 patients with COVID-19 and DM2, metformin reduced the rate of ICU admission and death by 2.9 and 5.5 times, respectively [32]. In the study by Do et al., patients with COVID-19 and DM2 taking metformin had a significantly lower incidence of AKI than patients taking other glucose-lowering drugs (1.7% vs. 3.3%, *p* = 0.021, respectively) [33] Luo et al., in their study, showed, that metformine significantly decreased in-hospital mortality in patients with COVID-19 and DM2. The death rate in patients taking metformine was 2.9% (*n* = 3) vs. 12.3 (*n* = 22) in patients, who did not take metformin, *p* = 0.01 [34]. In the work of Bell S et al., it was shown that metformin significantly increased 28-day survival in patients with AKI [35]. 

The pleiotropic effects of metformin, such as anti-inflammatory and antiviral actions, are discussed in addition to the hypoglycemic effect. The anti-inflammatory effect is mediated by AMPK modulation, which decreases the expression of proinflammatory genes, decreasing levels of the inflammatory markers in patients with and without DM2 [36,37]. Moreover, metformin has a favorable influence on the endothelium, preventing its damage and decreasing the risk of thrombosis by inhibiting the platelet activation factor [38]. Metformin is thought to reduce the activity of NETosis. In the study by Menegazzo et al., serum levels of NETosis biomarkers (cell-free dsDNA, histones, elastase, proteinase-3, neutrophils gelatinase-associated lipocalin (NGAL), and lactoferrin) were measured in 91 patients with pre-diabetes (*n* = 18), DM2 with bad glucose control (*n* = 73). It was shown that metformin significantly reduced serum levels of NETosis markers such as elastase, proteinase-3, histones, and double-strand DNA. Moreover, metformin prevented DNA release and membrane translocation of PKC-βII and activation of NADPH oxidase in neutrophils [39]. By AMPK activation, metformin also interferes with binding to the SARS-CoV-2 receptor ACE2 by its phosphorylation and also reduces the expression of proinflammatory genes by inhibiting the mTOR pathway [40]. 

Our study has limitations. The number of samples for the study of angiopoietin-1, neutrophil elastase 2, KIM-1, and membrane attack complex was relatively small. These parameters were assessed only at the time of admission. The dynamics of serum angiopoetin-1, ELA 2, KIM-1, and MAC levels, as well as urinary KIM-1 levels, were not evaluated. 

## 5. Conclusions

The clinical course of COVID-19 was complicated by AKI in 38% of cases. The independent risk factors of AKI in patients with COVID-19 are male sex, cardiovascular diseases, and chronic kidney disease, as well as high serum angiopoetin-1 levels and a decrease in blood lymphocyte count and fibrinogen levels. We propose the prognostic model of AKI development, which include the combination of serum levels of angiopoetin-1 and KIM-1 on admission. 

## Figures and Tables

**Figure 1 biomedicines-11-01246-f001:**
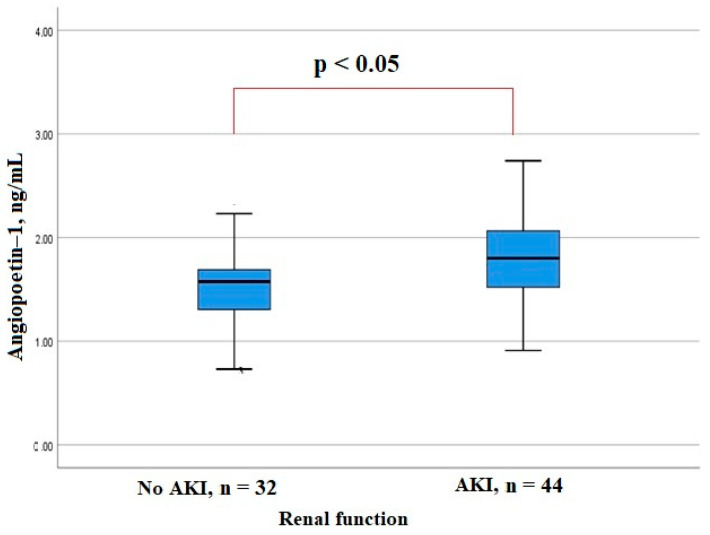
Serum angiopoetin-1 levels in patients hospitalized with COVID-19.

**Figure 2 biomedicines-11-01246-f002:**
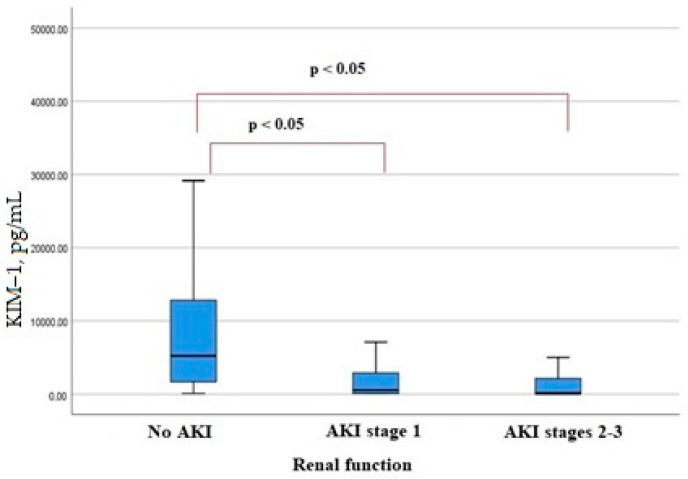
Serum KIM-1 levels in patients hospitalized with COVID-19.

**Figure 3 biomedicines-11-01246-f003:**
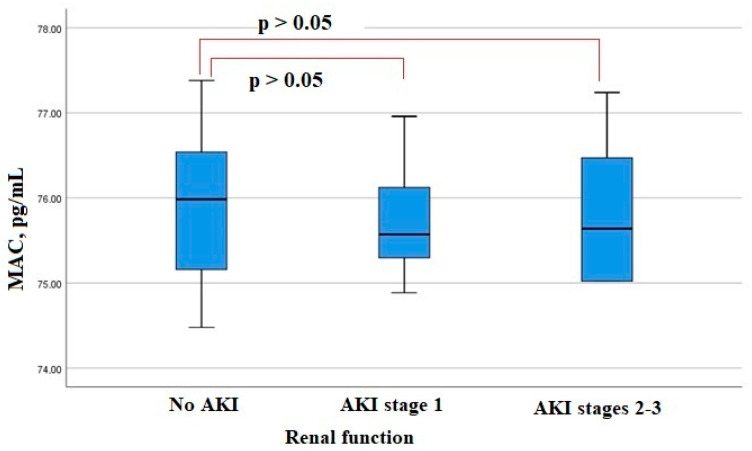
Serum MAC levels in patients hospitalized with COVID-19.

**Figure 4 biomedicines-11-01246-f004:**
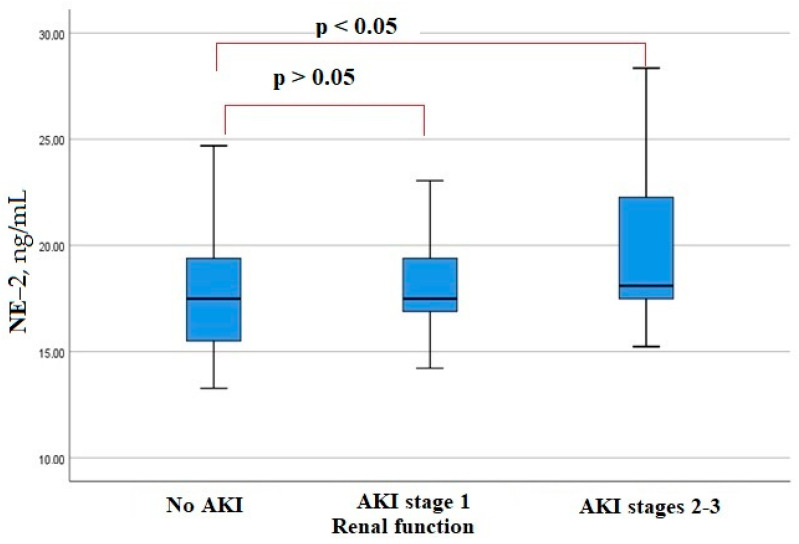
Serum neutrophil elastase 2 levels in patients hospitalized with COVID-19.

**Figure 5 biomedicines-11-01246-f005:**
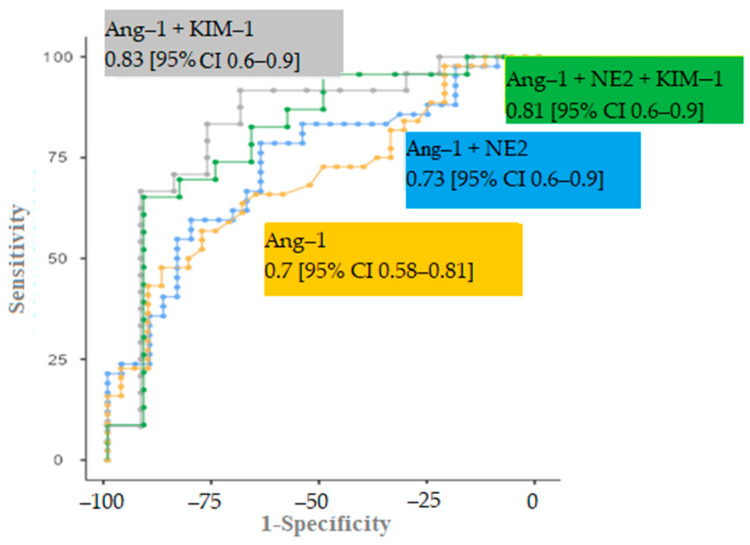
ROC curves of different markers combination models.

**Table 1 biomedicines-11-01246-t001:** Clinical and demographic characteristics of hospitalized patients with COVID-19.

Parameters	General Cohort (*n* = 500)	AKI Group (*n* = 190)	No AKI Group (*n* = 310)	*p*-Value
Age, years	73 [63; 80]	76 [65.75; 82]	71.5 [59; 79]	0.001
Men, n (%)	231 (46.2%)	104 (54.7%)	127 (41%)	0.003
Comorbidities:				
AH, n (%)	362 (72.4%)	151 (79.5%)	211 (68.1%)	0.006
DM, n (%)	124 (24.8%)	53 (27.9%)	71 (22.9%)	0.21
History of MI, n (%)	50 (10%)	29 (15.3%)	21 (6.8%)	0.002
AF, n (%)	74 (14.8%)	38 (20%)	36 (11.6%)	0.01
CKD C3-4, n (%)	117 (23.4%)	57 (30%)	60 (19.4%)	0.008
CHD III-IV FC, n (%)	47 (9.4%)	35 (18.4%)	12 (3.9%)	0.0001
BMI, kg/m^2^	28.4 [25.4; 32]	28.9 [25.8; 32.9]	28.1 [25.15; 31.7]	0.197
Obesity, n (%)	188 (37.6%)	78 (41.1%)	110 (35.5%)	0.154
Maximum area of lung damage on chest CT:				0.0001
No pneumonia, n (%)	5 (1%)	2 (1.1%)	3 (1%)	
Less than 25%, n (%)	98 (19.6%)	31 (16.3%)	67 (21.6%)	
25–50%, n (%)	191 (38.2%)	61 (32.1%)	130 (41.9%)	
51–75%, n (%)	150 (30%)	54 (28.4%)	81 (26.1%)	
More than 75%, n (%)	50 (10%)	11 (5.8%)	8 (2.6%)	
Oxygen saturation on admission, %	95 [92; 96]	94 [91; 96]	95 [93; 96]	0.024
AKI, n (%)	190 (38%)			
Stage 1, n (%)		151 (79.5%)		
Stage 2, n (%)		25 (13.1%)		
Stage 3, n (%)		14 (7.4%)		
Respiratory support				0.0001
Oxygen insufflation, n (%)	194 (38.8%)	74 (38.9%)	120 (38.7%)	
Mechanical ventilation:				
Non-invasive, n (%)	8 (1.6%)	8 (4.2%)	-	
Invasive, n (%)	50 (10%)	43 (22.6%)	7 (2.3%)	
Glucocorticoids, n (%),	357 (71.4%)	135 (71.1%)	222 (71.6%)	0.893
Antibiotics, n (%)	325 (65%)	157 (82.6%)	168 (54.2%)	0.0001
Beta-lactames/fluorochinolones, n (%)	285 (57%)	126 (25.2%)	159 (31.8%)	0.0001
Amikacine/Vancomycine/Colistine/, n (%)	40 (8%)	37 (19.5%)	3 (1%)	0.0001
Immunobiological drugs (tocilizumab, olokizumab, levilimab), n (%)	108 (21.6%)	55 (28.9%)	53 (17.1%)	0.002
Hydroxychloroquine, n (%)	83 (16.6%)	42 (22.1%)	41 (13.2%)	0.01
Azithromycine, n (%)	71 (14.2%)	35 (18.4%)	36 (11.6%)	0.01
Diuretics, n (%)	104 (20.8%)	50 (26.3%)	54 (17.4%)	0.02
iACE/ARB, n (%)	217 (43.4%)	79 (41.6%)	138 (44.5%)	0.461
Metformin, n (%)	67 (13.4%)	24 (12.6%)	43 (13.9%)	0.03
Inotropes, n (%)	33 (6.6%)	29 (15.3%)	4 (1.3%)	0.001
ICU admission, n (%)	76 (15.2%)	65 (34.2%)	11 (3.5%)	0.0001
Outcomes:				0.0001
Discharged, n (%)	416 (83.2%)	123 (64.7%)	293 (94.5%)	
Died, n (%)	84 (16.8%)	67 (35.3%)	17 (5.5%)	
Duration of hospitalization, days	12 [9; 15]	14 [10; 18]	11 [9; 14]	0.0001

**Table 2 biomedicines-11-01246-t002:** OR of AKI in patients hospitalized with COVID-19 (logistic regression models).

Factor	OR (95% CI)	*p*-Value	OR (95% CI)	*p*-Value
	Unifactorial logistic regression		Multifactorial logistic regression	
Male sex	1.7 (1.2–2.5)	0.003	2.1 (1.4–3.1)	0.0001
CHD III-IV FC	5.6 (2.8–11.2)	0.0001	4 (1.9–8.2)	0.0001
ELA2, ng/mL	1 (0.9–1.1)	0.313		
Ang-1, ng/mL	5.7 (1.7–19.1)	0.005	5.8 (1.7–20)	0.006
MAC, pg/mL	0.9 (0.9–1.1)	0.572		
KIM-1, pg/mL	0.56 (0.46–0.68)	0.038	1 (1–1)	0.036
Minimal lymphocytes	0.4 (0.3–0.7)	0.0001	0.005 (0–0.9)	0.049
Minimal fibrinogen	0.7 (0.6–0.8)	0.0001	0.2 (0.1–0.8)	0.002
Proteinuria	1.8 (1.1–2.9)	0.021	1.2 (1.1–1.4)	0.049
Antibiotics	3.9 (2.6–6.2)	0.0001	1.9 (1.1–3.3)	0.019
Metformin	0.6 (0.3–0.9)	0.022	0.4 (0.2–0.7)	0.002
Mechanical ventilation	2.6 (2–3.3)	0.0001	2.2 (1.7–2.9)	0.0001

**Table 3 biomedicines-11-01246-t003:** ROC analysis of AKI serum biomarkers (angiopoetin-1, KIM-1, neutrophil elastase 2).

Cut-Point	Sensitivity, %	Specificity, %	PPV (%)	NPV (%)	AUC (95% CI)
Ang-1					
1.66 ng/mL	63.6	68.6	73.7	57.9	0.69 (0.58–0.81)
KIM-1					
905.1 pg/mL	93.3	52	53.9	92.9	0.72 (0.51–0.85)
ELA-2					
20.075 ng/mL	53.4	84.6	60	78.6	0.67 (0.47–0.71)

## Data Availability

The datasets used during the current study are available from the corresponding author upon reasonable request.

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
