# Peer review of "Acute Kidney Injury in Hospitalized Patients with COVID-19: Risk Factors and Serum Biomarkers"

_biomedicines, 2023, doi:10.3390/biomedicines11051246_

Round 1
Reviewer 1 Report
This study found that the clinical course of COVID-19 was complicated by AKI in 38% of cases. The independent risk factors for AKI in patients with COVID-19 include male sex, cardiovascular diseases, chronic kidney disease, high serum angiopoetin-1 levels, decreased blood lymphocyte count, and decreased fibrinogen level. A prognostic model for AKI development was proposed, which includes a combination of serum levels of angiopoetin-1 and KIM-1 upon admission. The study appears to be logical and practical. We have some questions about its study:
Comments list
(Q.1) In the study design, AKI was defined according to the KDIGO Clinical Practice Guideline. Could you please provide more details on how to search for the ‘Baseline serum creatinine level’ in these hospitalized patients?
(Q.2) According to the results, several AKI biomarkers were examined. I am particularly interested in the finding that serum KIM-1 levels upon admission were lower in patients who developed AKI (Figure 2), which seems to slightly conflict with our expectations. As explained by the authors, active KIM-1 binding to the epitopes of the virus may temporarily decrease its amount in the blood, for example, immediately before the development of AKI. Therefore, KIM-1 levels may increase during serial follow-up over the course of the disease. Do you have any results on this?
(Q.3) According to the results, proteinuria also increased the risk of AKI in hospitalized COVID-19 patients. How was proteinuria defined and was it classified by severity for these patients?
Author Response
Dear Reviewer,
Thank you for your kind review of our manuscript.
Q1.1 In the study design, AKI was defined according to the KDIGO Clinical Practice Guideline. Could you please provide more details on how to search for the ‘Baseline serum creatinine level’ in these hospitalized patients?
In our study, we define the “baseline serum creatinine level” as patients’ serum creatinine level on admission. Thus, we defined AKI on the basis of serum creatinine changing >= 26.5 μmol/L from time of admission to peak follow-up value.
Q2.1 According to the results, several AKI biomarkers were examined. I am particularly interested in the finding that serum KIM-1 levels upon admission were lower in patients who developed AKI (Figure 2), which seems to slightly conflict with our expectations. As explained by the authors, active KIM-1 binding to the epitopes of the virus may temporarily decrease its amount in the blood, for example, immediately before the development of AKI. Therefore, KIM-1 levels may increase during serial follow-up over the course of the disease. Do you have any results on this?
Unfortunately, we didn’t evaluate the dynamics of serum KIM-1 levels in the late stages of disease as well as we didn’t evaluate urinary KIM-1 levels, so these findings demand further investigation.
Q3.1 According to the results, proteinuria also increased the risk of AKI in hospitalized COVID-19 patients. How was proteinuria defined and was it classified by severity for these patients?
We evaluated proteinuria in the spot urinalysis. It’s levels were not high and never reached nephrotic level. Therefore, we didn’t classify proteinuria by severity.
Best regards,
Authors
Reviewer 2 Report
This is a well planned, executed and presented study.
Introduction
“the level of acute phase proteins - CRP, 66 ferritin, LDH, the number of lymphocytes and peripheral blood platelets, procalciotonin, 67 D-dimer, fibrinogen, serum creatinine, glycose, electrolytes and proteinuria”
[please state from the literature why you chose these markers. What other markers should you have considered?]
Sample
“89 patients were selected from the main cohort to assess additional risk factors 70 for AKI in COVID-19”
[please state why you selected these patients and not others - how does this affect the generalisability of your findings?]
Discussion
“According to our findings, Metformin had protective effect on AKI and an unfa- 287 vorable outcome, reducing the risk of AKI by 2.7 times. These data are consistent with 288 data from other studies on COVID-19. For example, in the works of Chen Y et al., Cheng 289 et al., Gao et al. Metformin also reduced the risk of death, the frequency of ICU admis- 290 sion and the development of ARDS in patients with COVID-19.[15,29,20] In the work of 291 Bell S et. al, it was shown that Metformin significantly increased 28-day survival in pa- 292 tients with AKI.[31] In the literature, in addition to the hypoglycemic effect of Metfor- 293 min, its anti-inflammatory and antiviral properties are discussed.[32,33] Metformin is 294 thought to reduce the activity of NETosis.[34] Metformin interferes with binding to the 295 SARS-CoV-2 receptor and also reduces the expression of pro-inflammatory genes.[35] In 296 addition to the anti-inflammatory effect, the pleiotropic effect of metformin is discussed, 297 including a positive effect on the vascular endothelium and the correction of endothelial 298 dysfunction.”
[This one of the most interesting sections of your article. Please discuss the role of metformin in detail and present detailed numerical data from relevant studies].
Abstract
“High serum angiopoetin-1 levels, decrease of blood lymphocytes count and fibrinogen level 21 also increased the risk of AKI. Conclusions. AKI is an independent risk factor for death in patients 22 with COVID-19. We propose the prognostic model of AKI development, which include the combi- 23 nation of serum levels of angiopoetin-1 and KIM-1 on admission. Our model can help to prevent 24 AKI development and decrease mortality rate in patients with coronavirus disease.”
Conclusions
“According to the ROC analysis, serum angiopoietin-1 and KIM-1 levels were the 283 most informative combination for predicting AKI development within a week from the 284 day of hospitalization, indicating a predominant activation of endothelial dysfunction 285 and the virus's consumption of KIM-1 receptor determinants.”
[Your abstract differs from your conclusions. How can your model prevent AKI development? How is this accomplished?]
Author Response
Dear Reviewer,
Thank you for your interest in our manuscript!
Q1.2 please state from the literature why you chose these markers. What other markers should you have considered?
We chose these markers in accordance with many studies dedicated to evaluation of severity and prognosis of the novel coronavirus infection. https://www.ncbi.nlm.nih.gov/pmc/articles/PMC7574722/ Ng et al discuss markers of hyperinflammation in patients with COVID-19 and AKI and refer to secondary hemophagocytic lymphohistiocytosis. Cardinal markers of this syndrome are IL-2,6,7, ferritin and some others, which associated with COVID-19 severity and AKI incidence (https://pubmed.ncbi.nlm.nih.gov/25480521/). Moreover, CRP, LDH, ferritin, fibrinogen and D-dimer levels were evaluated by Yildirim et al (https://www.ncbi.nlm.nih.gov/pmc/articles/PMC8014704/). Serum levels of these biomarkers were significantly higher in patients with AKI than in in patients with preserved renal function.
We did not include data for IL-6 because this marker was evaluated only in patients with severe COVID-19, which could lead to biased conclusions.
Some explanations have been included in the manuscript.
Q2.2 please state why you selected these patients and not others - how does this affect the generalisability of your findings?
A special group of 89 patients included 47 patients who developed AKI during hospitalization and 42 patients without AKI. These subgroups were identified by gender, age, lung detection, incidence of CKD, and other comorbidities (DM, history of MI, CHF).
Q3.2 This one of the most interesting sections of your article. Please discuss the role of metformin in detail and present detailed numerical data from relevant studies
According to our findings, Metformin had protective effect on AKI and an unfavorable outcome, reducing the risk of AKI by 2.7 times and risk of death by 3.2 times. These data are consistent with data from other studies on COVID-19. For example, in our study of 136 with COVID-19 and DM2, Metformin reduced the rate of ICU admission and death by 2.9 and 5.5 times respectively. (https://erj.ersjournals.com/content/58/suppl_65/PA3653) In the study by Do et al patients with COVID-19 and DM2 taking Metformin had significantly lower incidence of AKI than patients taking ither glucose-lowering drugs (1.7% vs 3.3%, p=0.021, respectively). The limitation of this study was a small sample. (10.1016/j.diabet.2020.10.006) Luo et al in their study showed, that Metformine significantly decreased in-hospital mortality in patients with COVID-19 and DM2. The death rate in patients taking Metformine was 2.9% (n=3) vs 12.3 (n=22) in patients, who didn’t take Metformin, p=0.01. (https://www.ncbi.nlm.nih.gov/pmc/articles/PMC7356425/) In the work of Bell S et. al, it was shown that Metformin significantly increased 28-day survival in patients with AKI.[31]
In the literature, in addition to the hypoglycemic effect of Metformin, its anti-inflammatory and antiviral properties are discussed. The antiinflamatory effect is mediated by AMPK modulation, which decreases expression of proinflammatory genes, decreasing levels of the inflammatory markers in patients with and without DM2. [32,33] Moreover, Metformin has a favorable influence on the endothelium, preventing it’s damage and decreasing the risk of thrombosis by inhibiting platelet activation factor. (10.1038/srep36222) Metformin is thought to reduce the activity of NETosis. In the study Menegazzo et al serum levels of NETosis biomarkers (cell-free dsDNA, histones, elastase, proteinase-3, neutrophils gelatinase-associated lipocalin (NGAL) and lactoferrin) were measured in 91 patients with pre-diabetes (n=18), DM2 with bad glucose control (n=73). It was shown, that Metformin significantly reduced serum levels of NETosis markers such as elastase, proteinase-3, histones and double strand DNA. Their study in vitro showed, that Metformin prevented DNA release and membrane translocation of PKC-βII and activation of NADPH oxidase in neutrophils. (https://link.springer.com/article/10.1007/s00592-018-1129-8) By AMPK activation Metformin also interferes with binding to the SARS-CoV-2 receptor ACE2 by it’s phosphorillation and also reduces the expression of pro-inflammatory genes by inhibiting the mTOR pathway.[35]
The additional data has been included in the manuscript.
Q4.2 our abstract differs from your conclusions. How can your model prevent AKI development? How is this accomplished?
By using our model, we can define the group of patients with high risk of AKI and thus make clinical decisions that could lower this risk. Using non-nephrotoxic antibiotics, adequate hydration and timely using of anti-inflammatory agents such as IL-6 inhibitors could help accomplish this goal. Our study was retrospective; therefore, we couldn’t use our findings in clinical practice.
Yours faithfully,
Authors
Round 2
Reviewer 2 Report
Thanks to the authors for their update of this excellent manuscript.